# UV-Activated NO_2_ Gas Sensing: Photoactivated Processes on the Surface of Metal Oxides

**DOI:** 10.3390/nano15231795

**Published:** 2025-11-28

**Authors:** Pavel Kutukov, Daria Kurtina, Sergey Maksimov, Marina Rumyantseva

**Affiliations:** 1Chemistry Department, Moscow State University, 119991 Moscow, Russia; 2Federal Research Center Kazan Scientific Center RAS, 420111 Kazan, Russia

**Keywords:** gas sensors, photoactivation, mechanism, mass spectrometry, DRIFTS, nanomaterials

## Abstract

In recent years, wide bandgap metal oxide semiconductors have become the base materials of choice for semiconductor gas sensor design. In this work, nanocrystalline ZnO, In_2_O_3_, and SnO_2_ were investigated when detecting NO_2_ under UV-photoactivation conditions. The materials were characterized by XRD, low-temperature nitrogen adsorption, and electron microscopy. The article considers the mechanism of sensor signal formation, as well as the mechanism of action of UV-light photoactivation, using an in situ multi-method approach. In situ mass spectrometry and in situ TR-DRIFTS were employed to study the impact of UV-light photoactivation on target gas adsorption equilibrium as well as the electrical and gas-sensing properties of the materials.

## 1. Introduction

Metal oxides ZnO, In_2_O_3_, and SnO_2_ are n-type wide-bandgap semiconductors with oxygen non-stoichiometry. The dependence of their surface composition, electrical conductivity, and work function on the composition of the gas phase above a sample is well known [1,2] and was first suggested as early as the 1950s for single-element semiconductors such as germanium [3]. Since then, semiconducting metal oxides (SMOXs) have become the backbone of semiconductor gas sensors [2,4,5], with considerable attention being given to the aspects of high surface area material synthesis and sensor element manufacturing [4,6,7,8], sensor response modeling [9,10,11], sensitivity enhancement, and selectivity tuning.

Despite being quite versatile, conventional SMOX-based materials for gas-sensing applications still possess some drawbacks. A majority of such materials require the sensor to be operated at elevated temperatures (300–500 °C) to achieve the proper sensor signal, response time, and lifetime values (“thermal activation”). For these purposes, semiconductor gas sensor devices usually incorporate a heater in the form of a nichrome string or a ceramic micro-hotplate [4,12,13]. Consequently, the operation of an uncovered electric heater element leads to high sensor power consumption and makes the sensors non-explosion-proof. A promising alternative to the thermal activation of the sensor response is photoactivation [14,15,16,17]. Photoactivation can be accomplished either by illuminating the material with short-wavelength light (from UV light to blue light) [15,18,19,20], or by first introducing photo-sensitive dopants (usually organic dyes), which permit the use of longer wavelengths (from green to red) [21,22,23]. Since nowadays UV LEDs have become readily available, the former implementation of photoactivation may be preferable due to lower manufacturing costs.

A large number of papers report that photoactivation results in an increase in the sensor signal of SMOX-based materials utilized to detect both reducing (mainly VOCs) and oxidizing (for example, NO_2_) gases [14,18,19,20,24,25]. However, in most cases, the scope of such studies is limited to the investigation of the sensor response itself, which is calculated as a function of the electrical conductivity (or resistivity) of the material, while the exact nature of the physico-chemical processes pertaining to the formation of said sensor signal is left out. Some studies do recognize the role of target gas adsorption equilibrium in sensor signal formation and thus do attempt to characterize the surface composition of the material under different conditions by means of DRIFTS, TR-DRIFTS, Raman spectroscopy, or XPS [24,26,27,28,29,30,31]. Frequently, however, the aforementioned methods are applied ex situ [24], and therefore, the results cannot definitively prove what processes are directly involved in sensor signal formation. True in situ or operando techniques suitable for the in-depth investigation of the sensor signal mechanism have emerged relatively recently and are being developed in the fields of both gas sensing and photocatalysis [29,31,32,33].

This work is an attempt to investigate the mechanism of sensor signal formation, as well as the mechanism of action of UV-light photoactivation, using an in situ multi-method approach. Nanocrystalline oxide materials ZnO, In_2_O_3_, and SnO_2_ were selected for this study based on their common n-type wide-bandgap semiconductor nature, their gas sensitivity when exposed to a wide range of target gases, and the variation in their electronic properties, particularly the equilibrium charge carrier concentration. A series of three samples was first characterized ex situ using XRD, specific surface area measurements (BET), and electron microscopy. Next, in situ mass spectrometry was utilized in conjunction with in situ TR-DRIFTS and gas-sensing measurements to try to prove which of the mechanisms of formation of the sensor response to NO_2_ suggested in the literature describe the underlying processes most accurately.

## 2. Materials and Methods

### 2.1. Synthesis of Nanocrystalline ZnO, In_2_O_3_, and SnO_2_

Nanocrystalline ZnO was synthesized by the thermal decomposition of zinc basic carbonate (CAS Number 5263-02-5, purum p.a., Sigma Aldrich, St. Louis, MO, USA) in air at 300 °C for 24 h according to the following reaction:Zn_5_(CO_3_)_2_(OH)_6_ → 5 ZnO + 2 CO_2_ + 3 H_2_O(1)

The resulting zinc oxide powder was allowed to cool down in air and then stored in a sealed container to prevent contamination by carbon dioxide or moisture.

Nanocrystalline indium (III) oxide was synthesized via a sol–gel route, starting with commercially available indium oxide. First, the commercially available In_2_O_3_ (CAS Number 1312-43-2, Sigma Aldrich, St. Louis, MO, USA; purified grade, >99.9%) was dissolved in a stoichiometric amount of 10% hydrochloric acid (CAS Number 1312-43-2, Sigma Aldrich, St. Louis, MO, USA; purified grade) plus a 5% excess, using magnetic stirring at 80 °C under reflux (2).In_2_O_3_ + 6 HCl → 2 InCl_3_ + 3 H_2_O(2)

The indium chloride solution was then diluted with distilled water (to a target concentration of 100 g/L) and then carefully titrated with an aqueous ammonia solution (CAS Number 7647-01-0, Sigma Aldrich, St. Louis, MO, USA; 25%, purified grade) until the pH reached 5. During this process, a gel consisting of indium hydroxide and hydrated indium oxides was formed (3).2 InCl_3_ + 6 NH_3_·H_2_O → In_2_O_3_·xH_2_O↓ + 6 NH_4_Cl + (6 − x) H_2_O(3)

The synthesis of nanocrystalline tin (IV) oxide was also accomplished via a sol–gel route, commencing with commercially available tin tetrachloride pentahydrate (CAS Number 10026-06-9, Sigma Aldrich, St. Louis, MO, USA). First, tin (IV) chloride pentahydrate was dissolved in distilled water to obtain a 100 g/L solution. The tin tetrachloride solution was then titrated with an aqueous ammonia solution (25%, purified grade) until the pH reached 5. In the course of the process, a gel composed of stannic acid was formed (4).SnCl_4_ · 5H_2_O + 4 NH_3_ · H_2_O → SnO_2_ · xH_2_O↓ + 4 NH_4_Cl + (9 − x) H_2_O(4)

Next, the In_2_O_3_·xH_2_O or SnO_2_·xH_2_O gel was repeatedly washed with distilled and deionized water, using centrifugation to separate the gel from the solution. The washing steps continued until the supernatant passed the silver chloride test (using AgNO_3_). The gel was then dried at 60 °C in a drying oven, and the resulting xerogel was ground in an agate mortar into a fine powder. The powder was subsequently annealed in a furnace at 500 °C for 24 h, forming nanocrystalline In_2_O_3_ or SnO_2_.

### 2.2. Materials Characterization

The phase composition and crystal structure of the synthesized oxides were studied by powder X-ray diffraction (XRD) with a DRON-4-07 diffractometer using CuKα radiation (wavelength *λ* = 1.54059 Å). The average crystallite size *D* was calculated using the Sherrer equation:(5)$$D\;=\;\frac{k\lambda }{\beta cos\Theta }$$
where *λ* is the wavelength of the X-ray radiation, nm; *β* is the full width at half maximum (FWHM) of a diffraction peak, rad; *θ* is the diffraction angle; and *k* is a coefficient equal to 0.9. The peak width was corrected for instrumental broadening using the following equation:(6)$$\beta \;=\;\sqrt{\beta_{exp}^{2}-\beta_{app}^{2}}$$
where *β_exp_* is the observed peak width at half height and *β_app_* is the instrumental broadening, which was taken as 0.09°. The three most prominent reflections for each oxide were used for the calculations.

The elemental composition of the materials was confirmed by X-ray fluorescence (XRF) analysis using a M1 Mistral (Bruker) micro-X-ray spectrometer.

The specific surface area of the nanocrystalline oxides was measured using the low-temperature nitrogen adsorption method and the BET adsorption isotherm. The experimental procedure was carried out using a Chemisorb 2750 instrument (Micromeritics). The relative error of specific surface area determination was 5%.

The morphology of the materials was studied by means of scanning (SEM) and transmission (TEM) electron microscopy, using Prisma E (Thermo Scientific, Brno, Czech Republic) and JEM-2100 (JEOL, Tokyo, Japan) instruments.

### 2.3. Gas Sensing Measurements

The experimental setup involves miniature gas sensor devices with embedded microheaters and is described in detail in [32]. In brief, the devices were constructed using a 1.5 × 1.5 mm Al_2_O_3_ ceramic substrate suspended in air on 4 platinum wires that connected to its top and bottom layers. The bottom layer was coated with a thin film of platinum to form an embedded micro-heater, while the top layer housed a pattern of platinum electrodes for the sensing material to be applied onto. The temperature and electrical resistance of the sensors were controlled and logged by a custom instrument described previously in [34] (see Appendix A) or a compatible one.

Two types of gas sensing measurements were conducted.

The first type was used to provide the electrical conductivity profiles of the materials during mass spectrometry experiments. The data from both mass spectrometry and the corresponding gas-sensing experiments for a given material could then be unified on a single plot. In order to achieve this, a cylindrical chamber (V = 50 mL), made of glass and holding up to 4 gas sensors and a UV LED, was connected to the output of the flow cell used in in situ mass spectrometry experiments, and the latter experiments were repeated with an empty flow cell but with 3 sensors mounted in the sensor chamber. The LED inside the sensor chamber was activated synchronously with that of the flow cell.

The second type was utilized to provide data for the sensor signal calculations. These experiments feature cyclical NO_2_ concentration profiles, which first expose the sensors to dry purifier air, then to the target gas, then again to dry purified air, and so forth. The sensor signal values (S) can be calculated according to Formula (7).*S = (R_gas_/R_air_)* − 1(7)
where *R_gas_* is the resistance of the material in an atmosphere containing nitrogen dioxide, and *R_air_* is the resistance in dry purified air. Gas-sensing experiments of this type were conducted in a separate device as described in [32]. Briefly, the setup consisted of a measurement chamber (vertical cylinder shape, V = 50 mL, made of PTFE); a constant-current LED driver (I = 0.1 A); a gas mixing system; and a custom-made device for sensor heater control and sensing material resistance measurement. The sensors were placed at the bottom of the chamber, and a UV LED was mounted in the upper part of the chamber. The gases were mixed in real time using a source of purified dry air, a certified NO_2_ 100 ppm gas cylinder, and multiple mass-flow controllers (Bronkhorst High-Tech B.V., Ruurlo, The Netherlands). The resulting mixture was fed into the chamber inlet located in its top wall (the exhaust port was located at the bottom of the chamber). The UV LED, the sensors’ micro-heaters, and the mass-flow controllers were all simultaneously controlled by PC software according to the experiment plan.

### 2.4. In Situ TR-DRIFTS

The in situ time-resolved diffuse reflectance IR spectra were obtained using a Frontier FTIR spectrometer (Perkin Elmer, Shelton, CT, USA), which was fitted with a Pike DiffuseIR diffuse reflectance accessory (Pike Technologies). The DiffuseIR accessory includes a flow cell with a heater and can be equipped with a KBr window to allow simultaneous UV illumination of the sample and IR spectra acquisition. To make simultaneous IR spectra acquisition and UV illumination possible, a UV LED (Lite-ON LTPL-C034UVH365 or compatible) was mounted inside the accessory over the sample cell in such a way that it did not interfere with the IR beams (Figure 1). The UV LED was driven in constant-current mode at 120 mA. The precise central wavelength of the UV LED was determined to be 371 nm (span of 355–395 nm) using an OceanOptics SR-series dispersive spectrometer. In order to avoid UV interference and possible detector damage, a ZnSe window was inserted into the outgoing beam path (on the right side of the accessory).

The spectra acquisition range was set to 500–4000 cm^−1^ with 1 cm^−1^ resolution and an accumulation value of 30 raw spectra per averaged spectrum (the acquisition of a single averaged spectrum takes approx. 2 min). The sample powder was placed in a sapphire crucible and flattened by pressing. The crucible was put into the heater inside the flow cell, and the cell was closed with a KBr window on top.

A typical in situ TR-DRIFTS experiment protocol is similar to that of the mass spectrometry experiments and includes the following steps: sample preparation at 150 °C in dry air (100 mL/min) for 5 min, followed by slow cooling down to 30 °C; switching to a gas mixture containing 100 ppm NO_2_ (10–20 mL/min) and waiting for 60 min; and two UV illumination cycles each lasting 30 min (at 30 °C and 100 °C).

Kinetic curves derived from TR-DRIFTS data were calculated using custom software that implemented the trapezoidal integration algorithm with an endpoint-weighted linear baseline approximation. Since the exact extinction coefficients for most bands are not known, the raw peak areas were used as a measure of relative adsorbate concentration. It should be noted that these values can only be used to characterize how the adsorbate surface concentration evolves with time and cannot be used to compare the adsorbate surface concentrations to each other.

### 2.5. In Situ Mass Spectrometry

In situ photostimulated mass spectrometry studies were conducted in the previously described custom-made apparatus [32,35], consisting of a flat flow-reactor-type sample cell with top and bottom quartz windows. It allows simultaneous sample heating, UV illumination of the sample surface, and control over the gas mixture composition flowing over the sample powder. The gas flow inside the flow cell was modeled with the SimFlow CFD software (version 4.0) to ensure that the cell geometry creates a turbulent flow regime, thereby ensuring the CSTR mode of operation while minimizing excess volume effects such as unswept eddies (the results can be found in the Appendix A). The gas mixture exiting the cell was fed into an MS7-200D quadrupole mass spectrometer (equipped with an RGA-200 analyzer, Stanford Research Systems, Sunnyvale, CA, USA) over a heated PEEK capillary. The mass spectra acquisition was performed with CDEM enabled (gain 993). The ionizer settings were as follows: electron energy 70 eV, emission current 1.0 mA, focus voltage −90 V, ion energy 12 eV. The CDEM gain was accounted for during data processing. The noise floor (NF) was set to 3.

### 2.6. Long-Term NO_2_ Adsorption Measurements

The terms “prolonged NO_2_ adsorption” and “long-term NO_2_ adsorption” will be used to refer to adsorption processes that take well over 6 h (up to days or months) to reach saturation even when the total surface area of the sample is less than 1 m^2^ (which is the typical total surface area of a sample). Long-term NO_2_ adsorption can be characterized using in situ mass spectrometry and in situ TR-DRIFTS data, as the corresponding experiments typically last from 6 to 12 h, while providing information about the NO_2_ adsorption rate and surface concentrations of the NO_2_ adsorbate species, respectively.

Additionally, for ZnO, we employed an independent method for long-term NO_2_ adsorption measurement (CAPS spectroscopy) to confirm the conclusions that can be derived from in situ mass spectrometry and in situ TR-DRIFTS data. The experiment was conducted in the same flow cell that was used in the TR-DRIFTS experiments. The output of the flow cell was connected to a Teledyne API N500 CAPS NO_x_ Analyzer. The data were collected continuously over a Modbus-TCP interface. It was established that at a total flow rate of 100 mL/min of gas mixture through the flow cell and 1 L/min through the analyzer (diluted with purified air), rapid removal and insertion of the sample did not introduce artifacts into the measurement due to the large time constant of the analyzer.

## 3. Results

### 3.1. Phase Composition and Morphology

#### 3.1.1. Zinc Oxide

The powder diffraction pattern of the nanocrystalline ZnO is shown in Figure 2. The sample contains only a single phase with all of the reflections belonging to the wurtzite (hexagonal) phase of zinc oxide. The average crystallite size, as calculated using Formula (5), is 15 nm. As reported previously, zinc oxide crystallites exhibit anisotropy [32,35] and grow as needles or plates (Figure 3) with a characteristic length of ~100 nm and a characteristic thickness of 10 to 20 nm. Those needles and plates tend to form larger porous agglomerates. According to the BET model, the specific surface area of the nanocrystalline ZnO is about 4 m^2^/g.

TEM imaging revealed that the anisotropic structures of the nanocrystalline ZnO are composed of multiple grains and break apart upon ultrasonication, which is involved in the TEM sample preparation process (Figure 3b). This is consistent with literature data on the ultrasonic treatment of ZnO nanoparticles [36]. Since the needles and plates observed by SEM comprise multiple loosely bound nearly spherical grains, such anisotropy is unlikely to cause significant changes to the electronic properties of the semiconductor. The grain size distribution calculated based on TEM data (Figure 3c) is centered at 25 nm (SD = 18 nm).

The choice of a different annealing temperature (300 °C) for ZnO was made intentionally to produce nanocrystalline zinc oxide that has a comparable mean particle size and specific surface area to that of nanocrystalline indium and tin oxides annealed at 500 °C. It is well known that higher annealing temperatures produce material with a significantly larger mean particle size and a lower specific surface area [37]. Comparable morphology makes differences caused by chemical composition and electronic properties stand out, simplifying the interpretation of experimental data.

#### 3.1.2. Indium Oxide

The powder diffraction pattern of the nanocrystalline In_2_O_3_ is shown in Figure 4. All of the reflections belong to the bixbyite (cubic) phase of indium oxide. The average crystallite size, as calculated using Formula (5), is 22 nm. According to the BET model, the specific surface area of the nanocrystalline In_2_O_3_ was determined to be 10 m^2^/g.

According to TEM imaging (Figure 5), the shape of the indium oxide nanoparticles is mostly spherical with particle size varying from 5 to 33 nm. Occasional rod-like structures having a characteristic length of 30–60 nm were also observed.

#### 3.1.3. Tin Oxide

The powder diffraction pattern of the nanocrystalline SnO_2_ is shown in Figure 6. All of the reflections belong to the cassiterite (tetragonal) phase of tin oxide. The average crystallite size, as calculated using Formula (5), is 11 nm. According to the BET model, the specific surface area of the nanocrystalline SnO_2_ was calculated to be 25 m^2^/g.

TEM imaging (Figure 7) revealed that the particle size of the nanocrystalline SnO_2_ varies from 6 to 34 nm with a relatively narrow size distribution (SD = 5 nm) centered at 16 nm.

### 3.2. Photostimulated Processes on Pure Oxides

Photostimulated processes taking place on the surfaces of the oxide materials were first studied by in situ mass spectrometry in an atmosphere of high-purity helium mixed with 100 ppm of nitrogen dioxide. To demonstrate the effect of UV illumination of the samples, the experiment was designed as follows. First, the gas mixture was passed through the cell under dark conditions at RT (room temperature) for 2 h to achieve a steady state at 30 °C. Next, the UV light was turned on for a period of 30 min; then, the UV light was switched off, and the sample was allowed to rest for 30 min. Afterwards, the cycle was repeated at 50 °C and 100 °C with the time taken to achieve a steady state at a new temperature setting being shortened to 30–45 min. Concentrations of NO_2_ and O_2_ in the gas mixture leaving the sample cell correspond to ion currents for ions with m/z = 30 (NO^+^ fragment) and 32 (O_2_^+^), respectively. An ion current at m/z = 46 (NO_2_^+^) can also correspond to the NO_2_ concentration. However, in our setup, its intensity was found to be significantly lower than that of the signal at m/z = 30, which was due to the significant NO_2_ fragmentation that occurs when operating the ionizer at factory default settings. Furthermore, NO_2_^+^ is not the only ion capable of generating an ion current at m/z = 46. Particularly in the case of ZnO, adsorbed carbon dioxide can also generate ion currents at m/z = 12, 22, 44, 45, and 46, obscuring the NO_2_^+^ due to the relatively high natural abundance of the isotopes ^18^O, ^17^O, and ^13^C. It can also be noted that adding nitrogen to the helium carrier gas leads to the direct UV-activated photolysis of NO_2_, possibly by enabling the last step of the photolysis that requires the activated complex to be struck by another particle with sufficient energy [38]. In pure helium with an empty sample cell, no direct photolysis was observed. Conductivity data were obtained separately, albeit under the same experimental conditions, and then overlaid onto the mass spectrometry data.

#### 3.2.1. Zinc Oxide

In situ mass spectrometry data concerning the nanocrystalline ZnO were discussed in detail in our previous work [32]. In short, UV illumination of ZnO causes an apparent irreversible photoadsorption of both nitrogen dioxide and oxygen. However, it is crucial to emphasize that this apparent photoadsorption, as seen in ion current plots, does not necessarily represent a reversal of adsorption equilibrium (compared to that in dark conditions). If the material exhibits prolonged NO_2_ adsorption in dark conditions (or even absorption by means of a chemical reaction), then an increase in the adsorption reaction rate will produce the same results. While mass spectrometry experiments can be utilized to detect long-term NO_2_ adsorption under dark conditions (since their total duration is around 12 h), NO_2_ fragmentation issues initially impeded any decisive interpretation of the results. Therefore, an independent method, making use of CAPS spectroscopy, was also applied to this task (in addition to in situ mass spectrometry): a crucible containing ZnO was placed into a flow cell fed with 100 ppm of NO_2_ in N_2_, and the sample was left to adsorb NO_2_ for 6 h. Afterwards, the crucible was quickly extracted from the cell and then reintroduced again; the cycle was repeated twice (Figure 8). The output of the cell was continuously monitored by a CAPS NO_x_ analyzer. The difference between NO_2_ concentrations at the cell’s inlet and at its output (ΔC) is directly related to the adsorption reaction rate according to the mass conservation law; therefore, the results clearly show that nanocrystalline ZnO indeed is capable of prolonged NO_2_ adsorption (this point will be explained further in the discussion). Because different flow cells were used for mass spectrometry and CAPS spectroscopy measurements, the influence of NO_2_ adsorption on the exact NO_2_ concentration inside the cell’s outlet is less pronounced in Figure 8 than in the mass spectrometry results.

In situ TR-DRIFT spectroscopy data are presented as differential spectra [39]. This means that after each experimental stage, a new baseline was acquired, and the spectrum for the subsequent stage was recorded relative to this new baseline. This way, each curve in the TR-DRIFTS plots represents only the changes that happened during a selected experimental stage. Full, chronological TR-DRIFT spectra are provided in the Appendix A.

In situ TR-DRIFTS data confirm that pure nanocrystalline ZnO adsorbs NO_2_; room-temperature TR-DRIFT spectra are presented in Figure 9. The exposure of ZnO to nitrogen dioxide leads to the formation of a plethora of adsorbates (Table 1), including mono- and bidentate nitrates, nitrites, and possibly nitro groups (N-coordinated NO_2_^−^). Surface hydroxylation (3400–3700 cm^−1^) can also be observed, while the broadened band at ~3100 cm^−1^ turns negative, which suggests a dissociation of adsorbed water [40]; however, these effects can also be a consequence of the subtle humidity changes caused by gas mixture switching.

It is possible to calculate kinetic curves for effective surface concentrations of adsorbates using in situ TR-DRIFTS results (Figure 10). Once again, the kinetic curves prove that ZnO demonstrates prolonged NO_2_ adsorption, but these data make it possible to discern what the product of prolonged adsorption is: namely, the monodentate nitrate (m-NO_3_^−^). At the same time, the surface concentration of nitrites and nitro groups quickly reaches saturation. As expected, UV illumination of nanocrystalline ZnO speeds up NO_2_ adsorption but, curiously, only the one resulting in bidentate nitrate (b-NO_3_^−^) formation. These data provide an experimental basis for the separation between “long-term” and “short-term” NO_2_ adsorption processes.

Temperature increase has two major consequences (Figure 11). First, most nitrate ions transition to bidentate or bridging coordination, as illustrated by the redshift of the absorption band ca. 1302 cm^−1^ and the blue shift of the absorption band ca. 1509 cm^−1^. Second, negative peaks at ca. 1433 cm^−1^ and ca. 1098 cm^−1^ in the differential spectrum indicate a partial thermal-activated desorption of surface nitrate and nitrite species.

#### 3.2.2. Indium Oxide

In situ mass spectrometry results for the nanocrystalline indium (III) oxide are presented in Figure 12. Indium oxide, like zinc oxide, contains some adsorbed carbon dioxide and exhibits CO_2_ thermal desorption and photodesorption. However, in this case, the latter produces weaker ion currents. Together with a significantly larger concentration of NO_2_ exiting the flow cell, represented by larger ion currents at m/z = 30 and 46, it becomes possible to ascribe most of the ion current at m/z = 46 to NO_2_^+^. Therefore, it can be stated that nanocrystalline In_2_O_3_ demonstrates an apparent photodesorption of NO_2_, which is probably the result of the adsorption reaction slowing down under UV illumination. Interestingly, at higher temperatures (100 °C), the shape of the curve corresponding to the ion current at m/z = 32, which is assigned to O_2_^+^, starts to closely resemble those of NO_2_ (m/z = 30, 46). It can be assumed that at elevated temperatures, a real desorption of adsorbates containing an excess of oxygen (like nitrates [31]) may take place or at least that the mechanism of the underlying process changes.

An increase in sample conductance under UV illumination occurs mainly due to the photogeneration of non-equilibrium charge carriers and subsides after the UV light is turned off because of recombination and, possibly, electron trapping by the continuous ionosorption of NO_2_ [26]. Increasing the temperature of the sample (from 30 °C to 50 °C) leads to an increase in the conductance in accordance with the Fermi–Dirac statistics of charge carriers in semiconductors [1]. Any further increase in temperature, though, leads to the opposite effect: a slight negative trend appears that can be explained by carrier lifetime reduction.

The in situ TR-DRIFT spectroscopy results for indium oxide are similar to those for zinc oxide: all possible adsorbates of NO_2_ can be found on the surface of indium oxide, including mono- and bidentate nitrates, nitrites, and possible nitro groups (Table 2). The differential spectra (Figure 13) corroborate that the apparent photodesorption observed in mass spectrometry data at room temperature is, in fact, due to a decrease in the adsorption rate. There is no evidence of desorption on *differential* spectra, i.e., there are no sharp “negative” peaks. Considering two differential spectra corresponding to periods of 5 and 60 min of exposure to an atmosphere containing NO_2_, it is possible to separate two stages of NO_2_ adsorption with the band at 1602 cm^−1^ being assigned to physisorbed NO_2_ based on its proximity to that of free NO_2_ (1612–1618 cm^−1^) [8,43]. The aforementioned band quickly starts to overlap with the higher-energy components of doublets belonging to nitrates vibrations; nevertheless, during the first minutes of the experiment, this band is a singlet and thus represents NO_2_. The slight redshift of the band, compared to gaseous NO_2_, is expected, since interactions with adsorbed water or hydroxyl layers lower the bond strengths within the molecule. A significant baseline drift after exposure to both NO_2_ and UV light can be explained as a change in free charge carrier concentration, resulting in altered reflectivity according to the Drude model [44].

TR-DRIFT spectra acquired at elevated temperatures (100 °C, Appendix A) show no signs of UV-activated desorption of any nitrogen dioxide adsorbates. Similarly to the case of ZnO, nitrate adsorbate species undergo a coordination change from monodentate to bidentate.

#### 3.2.3. Tin Oxide

The in situ mass spectrometry results for the nanocrystalline SnO_2_ are presented in Figure 14. The conductance plot does not match the ion current plots as closely as it did in previous experiments due to a small change in the mass spectrometric experiment timings (more time is allocated to relaxation after UV illumination and after heating). Once again, the oxide possesses carbon dioxide on its surface, resulting in an overlap of ion currents at m/z = 46. In this case, the overlap is significant, yet it is still possible to note a feature emerging at 50–100 °C that was not present for the other oxides. A pronounced decrease in ion current corresponding to the NO_2_^+^ at 100 °C can be observed despite the overlap. Simultaneously, there is an increase in the ion current corresponding to NO^+^, suggesting that the ion current generated by O_2_^+^ is probably balanced by two opposing processes: oxygen photoadsorption and NO_2_ photoactivated dissociation. Photodesorption of NO_2_ is also possible at 50 °C.

According to in situ TR-DRIFTS, tin oxide adsorbs nitrogen dioxide, forming surface nitrates and nitro groups (no nitrites were found in this case). The spectrum is similar to that of indium oxide discussed above (Appendix A). The band assignment is given in Table 3. The IR band broadening in the tin oxide spectra is smaller than in indium oxide spectra; therefore, it was possible to accurately calculate the kinetic curves (Figure 15). The latter once again show that prolonged nitrogen dioxide adsorption results in the accumulation of nitrates on the surface, while adsorption leading to nitro-group formation quickly reaches saturation. UV illumination causes the rate of the prolonged adsorption to decline and may lead to a small photodesorption (right after the UV light is turned on). Despite numerous attempts, TR-DRIFTS experiments at elevated temperatures (100 °C) were unsuccessful in detecting the photodesorption suggested by the mass spectrometry results.

As for the possible photo-dissociation, the TR-DRIFT spectra do not directly support this assumption. However, the presence of a photo-dissociation process does not necessarily require that its products appear in TR-DRIFT spectra. If the products undergo rapid desorption, their corresponding adsorbate species can be regarded as intermediates. The surface concentration of intermediates approaches zero, which makes their detection with DRIFTS tricky, especially when large amounts of other adsorbate species are present. Furthermore, previous spectroscopic studies revealed that one of the expected photo-dissociation products—ionosorbed oxygen—can only be detected on reduced SnO_2_ surfaces [45].

### 3.3. Gas-Sensing Measurements

Figure 16 shows the changes in electrical resistance of sensors cyclically exposed to 0.5–2 ppm of NO_2_ at 30 °C, 50 °C, and 100 °C, both in dark conditions and under constant UV illumination. The measurements were performed in dry purified air. At high concentrations of NO_2_ (>1 ppm), the resistance of all three materials increases upon exposure to nitrogen dioxide, which is a classic n-type semiconductor gas sensor response to oxidizing gases. The ionosorption of oxidizing gases on the surface of the oxides leads to band bending and, in turn, to a rise in resistivity [4,9,46]. Both UV activation and an increase in temperature cause the resistivity of the oxide material to drop by increasing the charge carrier concentration inside the conduction band, though the effect of UV illumination is far more pronounced. The variation in *R_air_* of the materials encompasses variations in multiple parameters, including layer thickness and mean particle size, and cannot be attributed to differences in the equilibrium charge carrier concentration alone. For this reason, the *R_air_* of SnO_2_ quickly approaches the range limit of the instrument upon exposure to NO_2_ and therefore cannot be accurately determined at room temperature.

The UV activation of nanocrystalline ZnO and In_2_O_3_ yields two distinct results (Figure 17). The sensor signal of ZnO plummets when UV activation is used, while that of In_2_O_3_ increases 3–10 fold (the UV effect grows with temperature). Despite the similarities between SnO_2_ and In_2_O_3_ observed in mass spectrometry and TR-DRIFTS experiments, the influence of UV activation on the sensor signal of SnO_2_ at 100 °C differs strikingly from its effect on the sensor signal of In_2_O_3_: the signal is reduced almost 100 times when UV activation is used.

An attempt to calculate sensor response (air-to-gas transition) and relaxation (gas-to-air transition) times according to the exponential decay Formula (8) revealed that in most cases, two different time constants are present (Appendix A). This treatment excludes an additional 3-minute-long transient period that occurs after each gas mixture switching, corresponding to the gas mixing inside the cell (the cell’s time constant is approx. 30 s). It was found that UV activation leads to the following changes: first, it consistently shortens relaxation times (both *τ_1_* and *τ_2_* decrease under UV illumination); second, it speeds up the sensor response of ZnO but slows down that of In_2_O_3_ and SnO_2_ (both *τ_1_* and *τ_2_* are affected). Although it is tempting to assign the *τ_2_* of the response process to the long-term nitrate formation process, the absolute values of the calculated time constant values should be treated with caution due to the strong influence of the range of the data selected for the regression on the results.(8)$$R\left(t\right)=R_{0}+R_{1}\exp\left(-\frac{t-t_{0}}{\tau_{1}}\right)+R_{2}\exp\left(-\frac{t-t_{0}}{\tau_{2}}\right),\;\;\tau_{1}\ll \tau_{2}$$

Equation (8) includes the following terms: *R*(*t*)—sensor resistance (Ω) as a function of time; *R*_0_, *R*_1_, *R*_2_, *t*_0_—regression variables; *τ*_1_, *τ*_2_—regression variables, representing time constants (s); *t*—experiment time (s).

## 4. Discussion

One of the classic [8,26,30,31,32] model representations of the sensor signal formation of SMOX-based materials used for detecting NO_2_ is the simple ionosorption with nitrite formation (9). The nitrite formation, albeit limited, was indeed observed in the TR-DRIFT spectra reported in this work, as well as in Raman spectra reported previously [27], thus supporting this model. Moreover, the kinetics of nitrite formation coincide with that of conductivity changes in the material: both the surface nitrite concentration and the conductivity changes caused by exposure to NO_2_ quickly reach a plateau, as opposed to the long-term nitrate formation.(9)$$NO_{2\left(gas\right)}+e_{\left(C.B.\right)}^{-}↔NO_{2\left(ads\right)}^{-}$$

This model suggests that gaseous NO_2_ is adsorbed on the surface of the material, trapping an electron from the conduction band. This trapping creates a surface charge that gives rise to an electric field, which penetrates the semiconductor’s bulk according to the Poisson–Boltzmann equation and the Debye length (10), leading to band bending [10].(10)$$\lambda_{D}=\sqrt{\frac{\epsilon \epsilon_{0}kT}{q^{2}n_{bulk}}}$$

Depending on the grain size and Debye length of the material, the porous sensing layer operates in either the flat bands mode or grain-boundary mode [9,11]. Therefore, it is a feasible suggestion that in dark conditions, nanocrystalline indium oxide material, with the highest *n_bulk_* out of all three materials, operates in grain-boundary mode, while the other two materials operate in the flat bands condition (especially ZnO). A transition from flat-bands mode to grain-boundary mode, arising from non-equilibrium carrier photogeneration, can (partially) explain the degradation of the sensor signal of ZnO and SnO_2_ when operated with UV activation. It is unclear whether the possible photodissociation of NO_2_ on the surface of SnO_2_, which was observed at 100 °C (Figure 14), contributed to the signal degradation, since process (11) leaves chemisorbed oxygen on the semiconductor surface, and oxygen desorption under UV illumination is unlikely because the oxides are known to exhibit oxygen photoadsorption [47,48].(11)$$2NO_{2\left(ads\right)}+e^{-}↔2NO_{\left(gas\right)}+O_{2\left(ads\right)}^{-}$$

Furthermore, regarding the influence of UV activation on the sensor signal, it should be noted that according to the Formula (7), non-equilibrium carrier photogeneration alone may be sufficient to increase the signal value due to the diminishing of *R_air_*.

Based on in situ TR-DRIFTS findings for In_2_O_3_ presented above, we can amend the mechanism (9) and include the physisorption stage (12), noting that the process of surface nitrite formation (accompanied by the electron trapping) takes measurable time.(12)$$NO_{2\left(gas\right)}↔NO_{2\left(ads\right)}$$

It is a widespread point of view [49,50,51,52] that UV activation enhances the properties of SMOX-based gas sensors employed for NO_2_ detection by causing a photodesorption of surface nitrites and thus releasing the trapped charge carriers back into the conduction band, relieving the surface charge and decreasing surface contamination. However, the combined in situ mass spectrometry and in situ TR-DRIFTS results prove that all three oxides are prone to prolonged NO_2_ adsorption at room temperature even when UV activation is used. Under the conditions considered in this work, UV activation at room temperature was found to be able to either promote or suppress the long-term adsorption rate of NO_2_ but not to reverse the direction of the process. Photodesorption can be observed only in mass spectrometry results for In_2_O_3_ and SnO_2_ and only at elevated temperatures (50–150 °C) together with the photoactivated dissociation of NO_2_. The lack of hypothesized photodesorption of the surface nitrite species is likely a consequence of the poor or even non-existent mobility of states belonging to the valence zone (holes) [1].

The fact that the long-term NO_2_ adsorption (with surface nitrate formation) demonstrates no matching significant long-term conductivity changes should not come as a surprise, because the surface charge density cannot surpass the Weisz limit [53]. This implies that nitrate formation does not require a charge transfer from the conduction band of the semiconductor to the adsorbate. Only a single model representation (13) of NO_2_ adsorption that satisfies the aforementioned condition (though on TiO_2_) can be found in the literature [54].(13)$$\left(2+n\right)NO_{2\left(ads\right)}+nO_{\left(lat\right)}^{2-}+e^{-}\to \left(1+n\right)NO_{3\left(ads\right)}^{-}+NO_{gas}+nV_{O}^{-}$$

The model first assumes that the dimerization of NO_2_ into N_2_O_4_, followed by nitrosyl nitrate formation (14), is thermodynamically favorable on the oxide surface (recently confirmed by DFT calculations [55]). Next, the nitrosyl cation reacts with the lattice oxygen (15) to form an adsorbed nitrogen dioxide molecule that can undergo dimerization (with any adjacent adsorbed NO_2_ molecule) again, continuing the process (13) in a chain-reaction manner. And every chain reaction has a termination reaction: in this case, the nitrosyl cation can be reduced to NO by the oxide (16), requiring the adsorption of a “replacement” NO_2_ molecule to continue the dimerization and the nitrate formation processes. Single-charged oxygen vacancies are known to be unstable in ZnO; they quickly undergo disproportionation according to reaction (16).(14)$$2NO_{2\left(ads\right)}↔ONONO_{2\left(ads\right)}↔NO_{\left(ads\right)}^{+}+NO_{3\left(ads\right)}^{-}$$(15)$${NO}_{\left(ads\right)}^{+}+O_{\left(lat\right)}^{2-}\to {NO}_{2\left(ads\right)}+V_{O}^{-}$$(16)$${NO}_{\left(ads\right)}^{+}+e^{-}\to {NO}_{\left(gas\right)}$$(17)$$2V_{O}^{-}\to V_{O}+V_{O}^{2-}$$

At first glance, the notion of a surface reaction leading to the formation of a number of oxygen vacancies might seem counterintuitive, but the experimental data actually support this model. The ion current value at m/z = 30 (fragment NO^+^) corresponding to 100 ppm of NO_2_ was calibrated to be 0.55 ± 10% pA using a certified gas cylinder. Comparing this value to the corresponding ion current values under dark conditions in Figure 12 and Figure 14, as well as to the data published previously [32], it is possible to plot the stationary concentration of NO_2_ determined at the output of the in situ mass spectrometry flow cell at different cell temperatures versus the metal–oxygen bond energy of the materials (Figure 18a). The stationary concentration of NO_2_ exiting the cell corresponds to its adsorption rate according to the mass conservation law, which can be written as (18):(18)$$\frac{dC}{dt}=-s\frac{d\theta }{dt}+\frac{w}{V}\left(C_{0}-C\left(t\right)\right)=0$$
where *C*(*t*) is the volume concentration of NO_2_ measured at the cells outlet, *s* is the effective area of the sample, *θ* is the surface coverage, *w* is the volumetric flow rate, *V* is the reactor volume, and *C*_0_ is the source concentration (100 ppm). Note that the first term of the sum represents the adsorption reaction rate. Hence, the deviation of the NO_2_ concentration (as measured at the outlet of the cell) from the source concentration can be expressed as (19):(19)$$\Delta C=-\frac{sV}{w}r_{ads}$$

The resulting plot (Figure 18a) demonstrates that the long-term adsorption rate decreases as the metal–oxygen bond energy increases, indicating that the reaction (15) indeed involves lattice oxygen. The temperature dependence of the adsorption rate follows the trend predicted by thermodynamics: the entropy change in adsorption (Δ_ads_S) must be negative; therefore, the system (provided that surface coverage is lower than its equilibrium value) approaches the equilibrium state when the temperature is increased, and thus the adsorption rate decreases.

At this point, one question remains unaddressed: why does UV activation speed up NO_2_ adsorption for ZnO but slows it down for In_2_O_3_ and SnO_2_? It is worth pointing out that the maximum photon energy of the UV LEDs used is 3.5 eV, which is lower than the *optical* bandgap energy of In_2_O_3_ and SnO_2_ (Figure 18b). Although the optical bandgap values of nanocrystalline materials can be lowered by the formation of “tails” of density of states (consisting of defect or surface states) near band edges [58], as well as by the symmetry implications of the nanocrystalline state, it is reasonable to consider the possibility of exciton generation in In_2_O_3_ and SnO_2_ under UV illumination [59,60]. Since excitons are neutral quasiparticles, their generation enables the non-equilibrium (photogenerated) charge carriers to reach the surface in spite of the presence of a large negative surface potential caused by ionosorption. Once the excitons reach the surface, they can participate in surface reactions, in particular in the reaction (16), increasing its rate and, in turn, slowing down the overall chain reaction of the surface nitrate formation (13). A decrease in the rate of a reaction that does not contribute to the sensor signal formation can have a beneficial effect on the gas-sensing properties of the material, giving rise to the sensor signal increase observed for UV-activated In_2_O_3_. In the case of ZnO, UV light almost certainly causes interband transitions to begin with, so no excitons can reach its surface. But the UV radiation can contribute to the nitrosyl nitrate formation step (14) through photo-excitation of the adsorbate molecules, ramping up the nitrate formation reaction rate and negatively impacting the sensor signal.

## 5. Conclusions

We have studied gas-sensor materials for NO_2_ detection based on ZnO, In_2_O_3_ and SnO_2_ using an in situ multi-method approach. For all the three materials, a significant irreversible contamination of the surface with nitrate species was observed during long-term operation in NO_2_-rich atmosphere at room temperature. We found no evidence of the photodesorption (promoted by photoactivation with UV light) of nitrogen dioxide adsorbates from the surface of the materials at room temperature. Still, under the aforementioned conditions, UV activation strongly impacts the sensor signal by controlling both electronic properties of the semiconductor (charge carrier density, porous layer conduction regime, exciton generation) and surface reaction rates. UV activation was found to suppress the sensor signal of ZnO and SnO_2_ but increase that of In_2_O_3_. Nevertheless, UV activation can be beneficial for both In_2_O_3_- and SnO_2_-based materials, since in both cases, it slows down sensor material degradation by reducing the rate of the reactions that lead to surface contamination (possibly through exciton generation). In addition, UV activation lowers the resistivity of the materials, driving down the complexity and cost of the measurement circuits. The impact of UV activation on the sensor kinetics is complex: on the one hand, it shortens the relaxation time; on the other hand, it can increase the response time (as observed for In_2_O_3_ and SnO_2_), creating a possible tradeoff between the response time and the material degradation rate. The presented in situ techniques open possibilities for the further investigation of NO_2_ adsorption kinetics in relation to the electrical properties of the materials under UV activation conditions.

## Figures and Tables

**Figure 1 nanomaterials-15-01795-f001:**
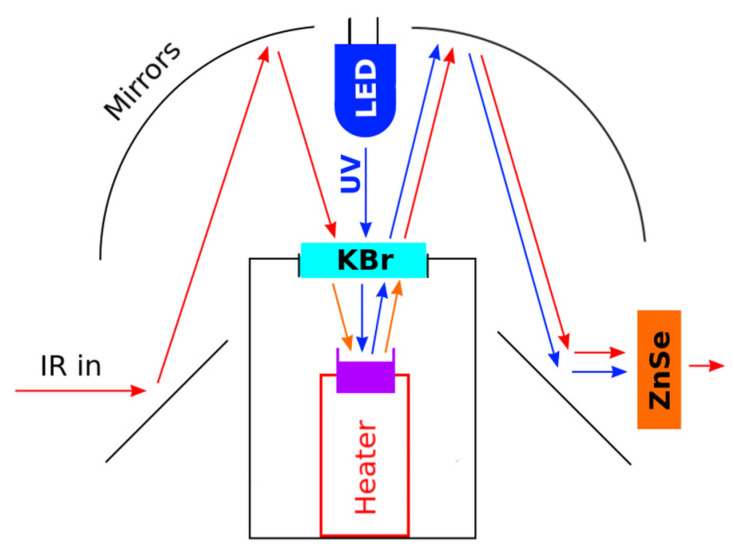
Schematic representation of the modified accessory for in situ TR-DRFITS. The sample crucible is painted purple. Both the heater and the sample are located inside the flow cell. Red arrows represent infrared light. Blue arrows represent ultraviolet light.

**Figure 2 nanomaterials-15-01795-f002:**
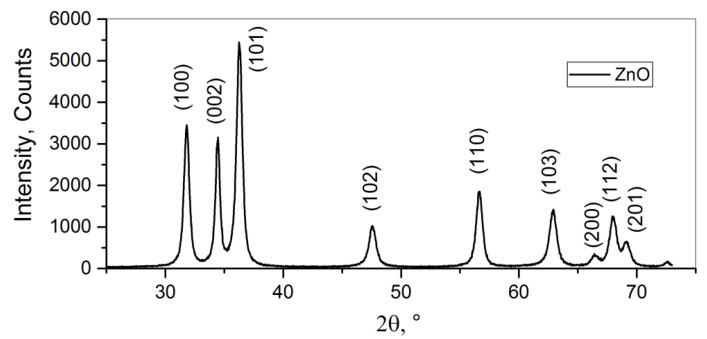
XRD pattern of the nanocrystalline ZnO.

**Figure 3 nanomaterials-15-01795-f003:**
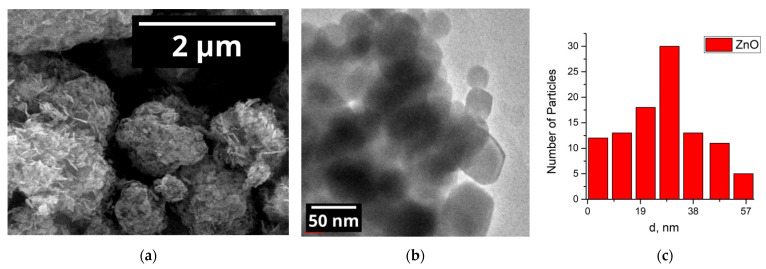
(**a**) SEM image of the nanocrystalline ZnO; (**b**) TEM image of the nanocrystalline ZnO; (**c**) particle size distribution, calculated based on the TEM data.

**Figure 4 nanomaterials-15-01795-f004:**
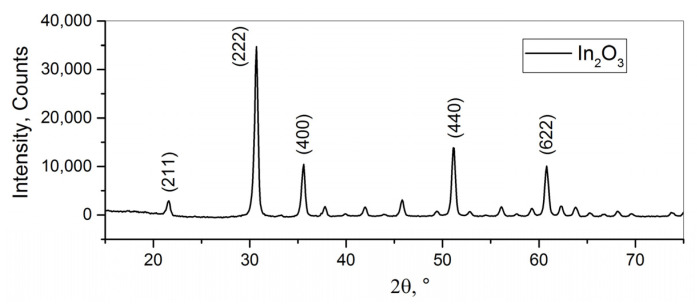
XRD pattern of the nanocrystalline indium (III) oxide.

**Figure 5 nanomaterials-15-01795-f005:**
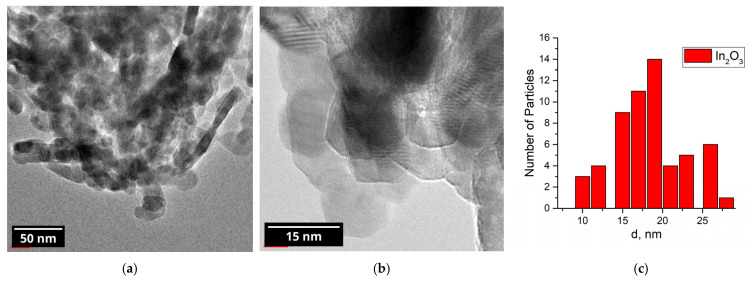
(**a**,**b**) TEM images of the nanocrystalline In_2_O_3_; (**c**) particle size distribution.

**Figure 6 nanomaterials-15-01795-f006:**
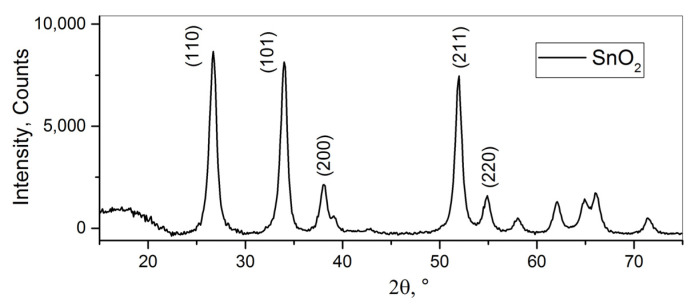
XRD pattern of the nanocrystalline tin (IV) oxide.

**Figure 7 nanomaterials-15-01795-f007:**
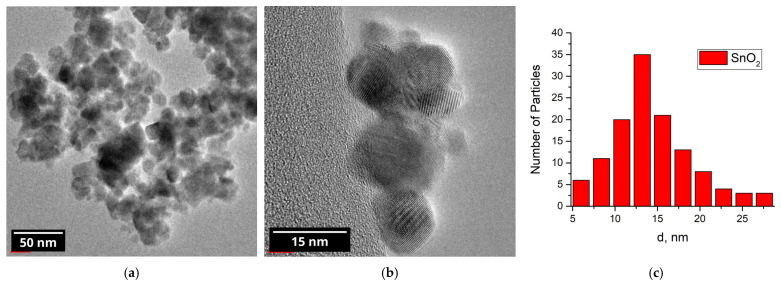
(**a**,**b**) TEM images of the nanocrystalline SnO_2_; (**c**) particle size distribution.

**Figure 8 nanomaterials-15-01795-f008:**
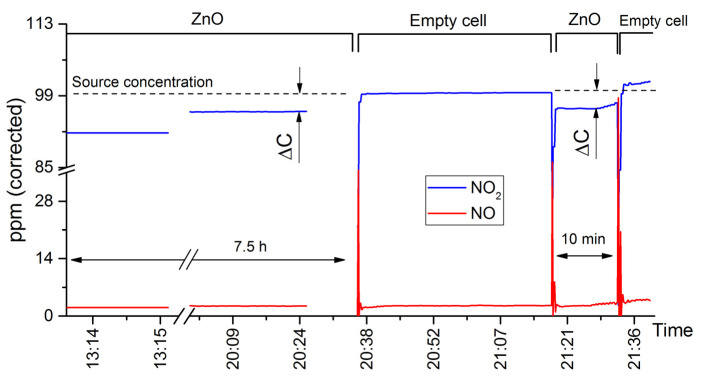
Long-term NO_2_ adsorption measurement (using CAPS NO_x_ analyzer) for the nanocrystalline ZnO at room temperature. The concentration scale was calibrated against a reference standard (certified 100 ppm NO_2_ gas mixture). NO concentration is at the instrument noise level. ΔC represents the difference between NO_2_ concentrations at the cell’s inlet and at its output.

**Figure 9 nanomaterials-15-01795-f009:**
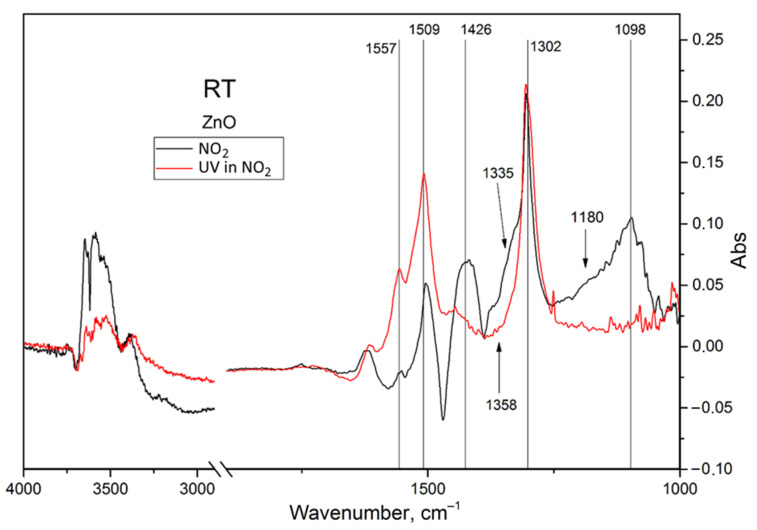
Differential TR-DRIFT spectra of the nanocrystalline ZnO exposed to 100 ppm of NO_2_ and then subjected to UV illumination in the same gas mixture (RT means “at room temperature”).

**Figure 10 nanomaterials-15-01795-f010:**
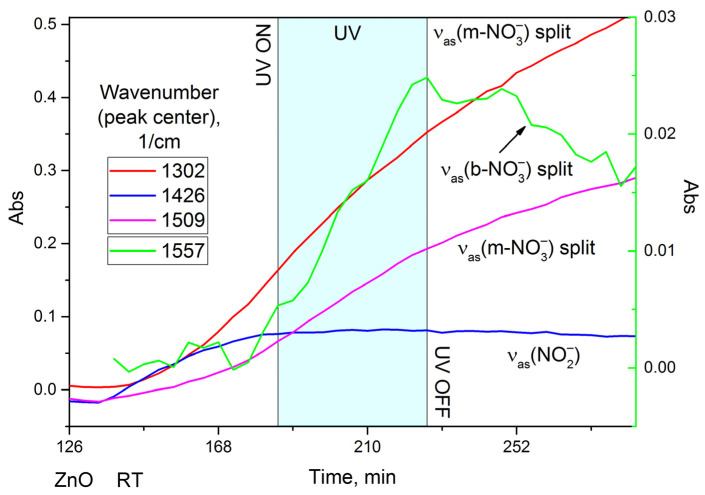
Kinetic curves calculated based on TR-DRIFTS data for nanocrystalline ZnO subjected to 100 ppm of NO_2_ and UV illumination (cyan region) at room temperature.

**Figure 11 nanomaterials-15-01795-f011:**
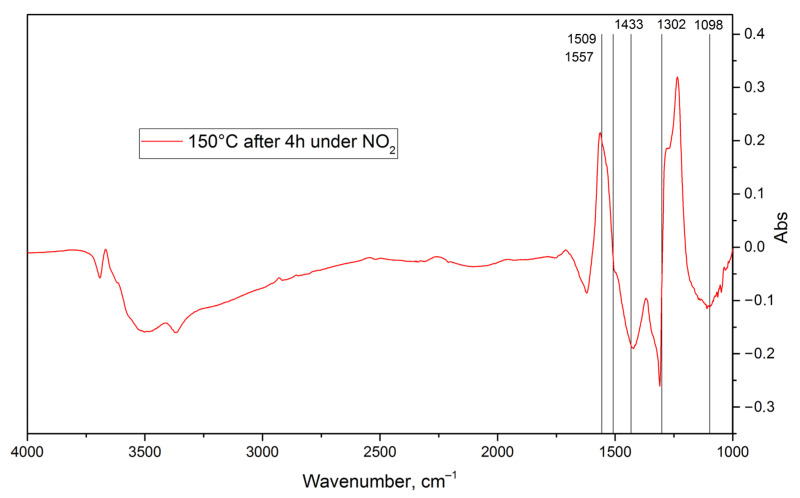
Differential TR-DRIFT spectrum of the nanocrystalline ZnO heated up to 150 °C after a 4-hour-long exposure to 100 ppm of NO_2_.

**Figure 12 nanomaterials-15-01795-f012:**
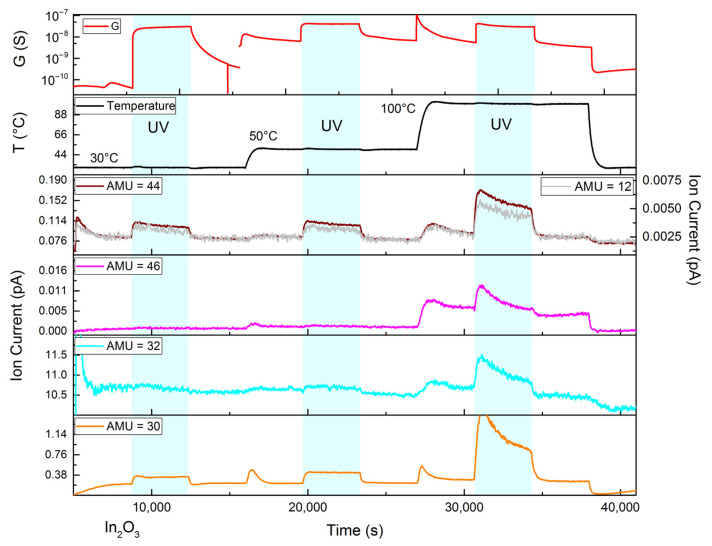
In situ mass spectrometry ion current plot for the nanocrystalline indium oxide subjected to 100 ppm NO_2_ in helium and UV illumination (cyan regions) at 30 °C, 50 °C and 100 °C. The curves for 12 and 44 m/z are geometrically similar, indicating that the ion current at 44 m/z indeed corresponds to CO_2_. NO_2_ at 46 m/z partially overlaps with CO^18^O. Top curve: electrical conductance (G, S = Ω^−1^) during a separate gas-sensing experiment (under the same conditions).

**Figure 13 nanomaterials-15-01795-f013:**
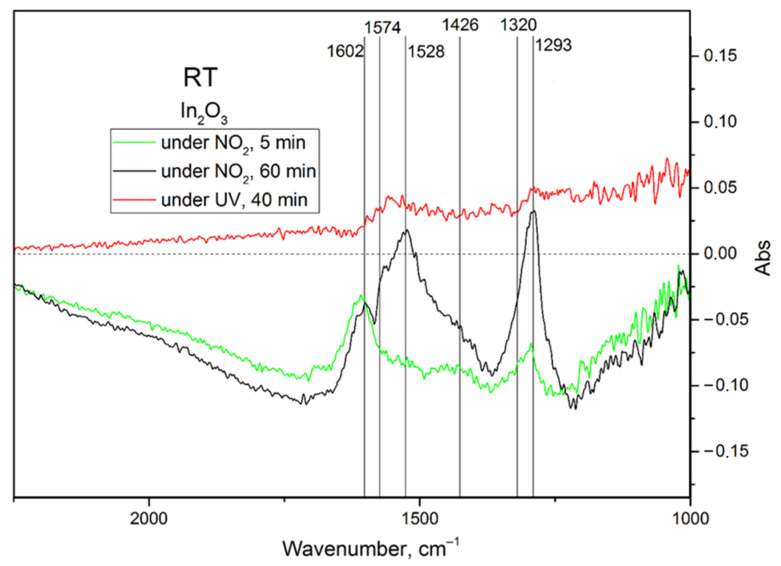
Differential TR-DRIFT spectra of the nanocrystalline In_2_O_3_ exposed to 100 ppm of NO_2_ and then subjected to UV illumination in the same gas mixture.

**Figure 14 nanomaterials-15-01795-f014:**
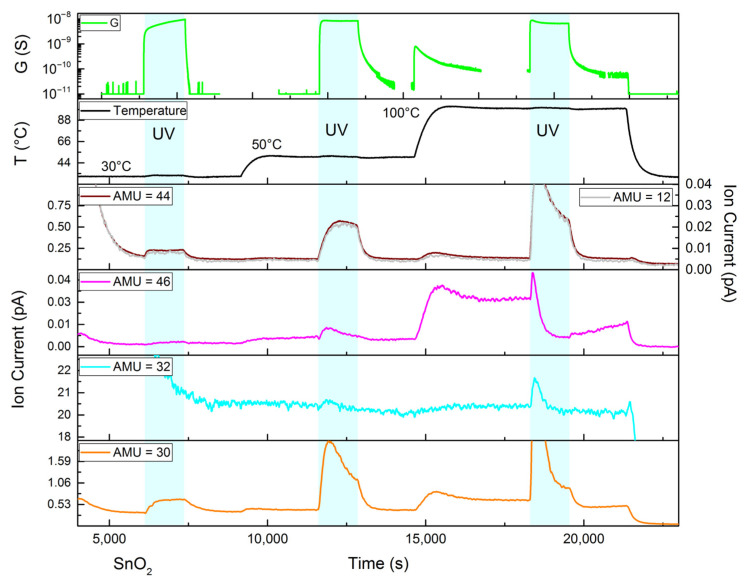
In situ mass spectrometry ion current plot for the nanocrystalline tin oxide subjected to 100 ppm NO_2_ in helium and UV illumination (cyan regions) at 30 °C, 50 °C and 100 °C. The curves for 12 and 44 m/z are geometrically similar, indicating that the ion current at 44 m/z indeed corresponds to CO_2_. NO_2_ at 46 m/z partially overlaps with CO^18^O. Top curve: electrical conductance (G, S = Ω^−1^) during a separate gas-sensing experiment (under the same conditions).

**Figure 15 nanomaterials-15-01795-f015:**
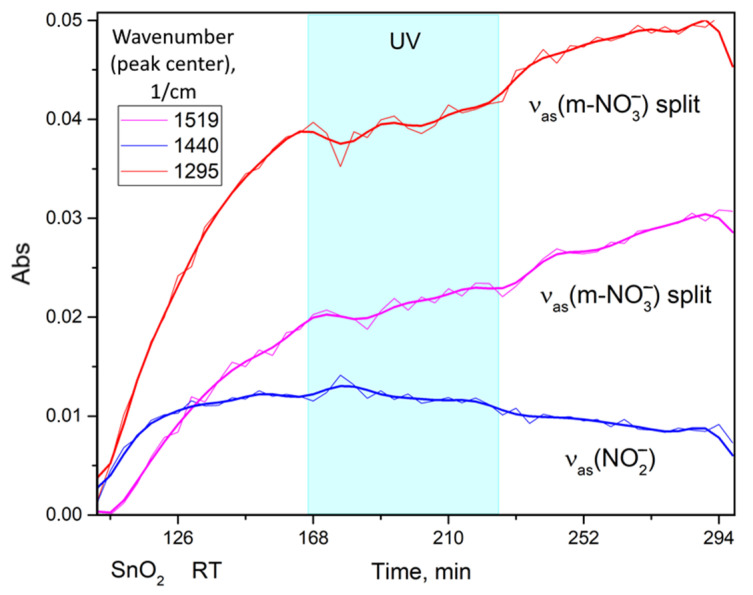
Kinetic curves calculated based on TR-DRIFTS data for the nanocrystalline SnO_2_ subjected to 100 ppm of NO_2_ and UV illumination (cyan region) at room temperature.

**Figure 16 nanomaterials-15-01795-f016:**
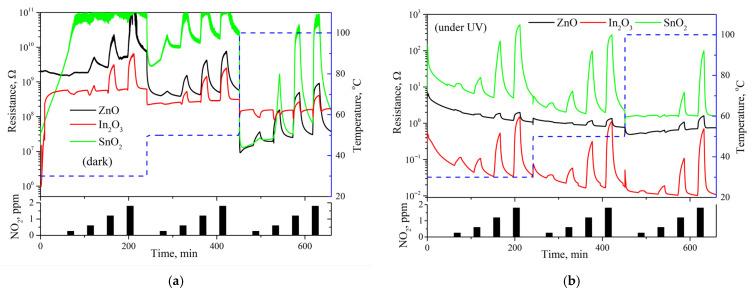
(**a**) Raw electrical resistance plots for the gas-sensing experiment conducted in dark conditions. The dashed line represents temperature. (**b**) Raw electrical resistance plots for the gas-sensing experiment conducted under UV illumination. The dashed line represents temperature.

**Figure 17 nanomaterials-15-01795-f017:**
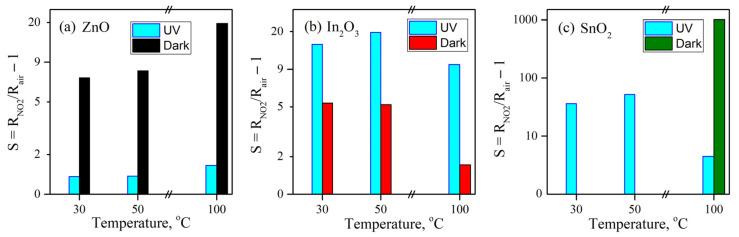
Calculated sensor signal values.

**Figure 18 nanomaterials-15-01795-f018:**
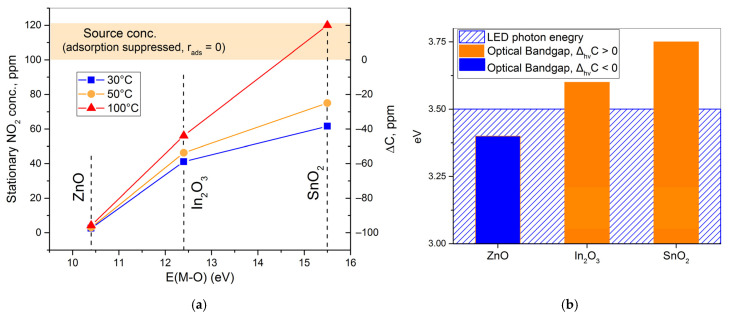
(**a**) Trend diagram for the stationary concentration of NO_2_ exiting the flow cell during in situ mass spectrometry experiments, metal–oxygen bond energy [26,48], and temperature. (**b**) Correlation diagram for *optical* bandgap energy [14,46,56,57] and UV photon energy.

**Table 1 nanomaterials-15-01795-t001:** Possible IR absorption band assignments for the nanocrystalline ZnO [8,41,42,43].

Band	Assignment
1098	$v_{s}$ $NO_{2}^{-}/NO_{3}^{-}\;$(characteristic of all symmetric N-O modes)
1180	$N,\;O\text{-}bidentate\;NO_{2}^{-}\;(N\text{-}O)$
1302–1294	$v_{as}\;NO_{3}^{-}\;split\;\left(Zn\text{-}O\text{-}N\;assym.\right)$
1335	$v_{s}\;NO_{2}^{-}\;nitro$
1358	$NO_{2}^{-}\;nitro\;$(frequently observed during decomposition of nitro-compounds)
1426	$v_{as}\;NO_{2}^{-}\;nitro$
1509	$v_{as}\;monodent.\;NO_{3}^{-}\;split\;(N=O\;assym.)$
1517–1530	$N,\;O\text{-}bidentate\;NO_{2}^{-}\;(N=O)$, $v_{as}\;monodent.\;NO_{3}^{-}\;split\;(N=O)$
1557	$v_{as}\;bidentate\;NO_{3}^{-}\;split\;(N=O)$

**Table 2 nanomaterials-15-01795-t002:** Possible IR absorption band assignments for the nanocrystalline In_2_O_3_ [41,42,43].

Band	Assignment
1240	$v_{s}\;N,O\text{-}bidentate\;NO_{2}^{-}\;\left(N=O\right)$
1290	$v_{as}\;NO_{3}^{-}\;split\;\left(Zn\text{-}O\text{-}N\;assym.\right)$
1320	$v_{s}\;NO_{2}^{-}\;nitro$
1426	$v_{as}\;NO_{2}^{-}\;nitro$
1528	$v_{as}\;N,\;O\text{-}bident.\;NO_{2}^{-}\;(N=O)$ $v_{as}\;monodent.\;NO_{3}^{-}\;split\;(N=O\;assym.)$
1553–1574	$v_{as}\;bident.\;NO_{3}^{-}\;split\;(N=O)$
1602	$v_{as}\;NO_{2}\left(ads.\right)$

**Table 3 nanomaterials-15-01795-t003:** Possible IR absorption band assignments for the nanocrystalline SnO_2_ [41,42,43].

Band	Assignment
1295	$v_{as}\;NO_{3}^{-}\;split\;\left(Me\text{-}O\text{-}N\;assym.\right)$
1335	$v_{s}\;NO_{2}^{-}\;nitro$
1440	$v_{as}\;NO_{2}^{-}\;nitro$
1519	$v_{as}\;monodent.\;NO_{3}^{-}\;split\;(N=O\;assym.)$
1578	$v_{as}\;bident.NO_{3}^{-}\;split\;(N=O)$

## Data Availability

The original contributions presented in this study are included in the article/Appendix A. Further inquiries can be directed to the corresponding authors.

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
