# Peer review of "UV-Activated NO2 Gas Sensing: Photoactivated Processes on the Surface of Metal Oxides"

_nanomaterials, 2025, doi:10.3390/nano15231795_

Round 1
Reviewer 1 Report
Comments and Suggestions for Authors
The material characterization, gas-sensing tests, and mechanism explanation in the manuscript show good correlation. However, the following issues need to be addressed:
- The introduction should add some current applications of in-situ TR-DRIFTS.
- There is a writing error on line 57 that needs correction.
- Figures 8 and 18 are not clear enough.
- The reason for selecting these three specific materials should be explained. Furthermore, since the morphologies of the three synthesized materials are different, the influence of morphology should be discussed in the text, as morphology significantly impacts the performance of MOS gas sensors.
- The description "The resulting plot (Figure 18a) demonstrates that long-term adsorption rate decreases as metal-oxygen bond energy increases, indicating that the reaction (14) indeed involves lattice oxygen" does not seem well-supported by Figure 18a. Furthermore, only the long-term NO2 adsorption measurement for ZnO is presented in the manuscript.
- Why are there no TEM images for ZnO? Additionally, the HRTEM images should have annotations marking the lattice fringes corresponding to the oxide.
Reviewer 2 Report
Comments and Suggestions for Authors
Please see the attachment.

Reviewer 3 Report
Comments and Suggestions for Authors
The authors study the mechanism of UV photoactivation of metal oxide-based sensors through in-situ characterization performed by mass spectroscopy and DRIFT analysis. In first part of the work they describe the preparation and characterization (by XRD, TEM, SEM and BET) of metallic oxides to use as sensor materials. They then report the characterization results obtained by mass spectroscopy and DRIFT analysis carried out in presence of NO2 gas at different temperatures under different conditions (in the dark before and after UV stimulation and during it), comparing them with the electric responses of the corresponding sensors. Finally, based on the results, they discuss different behaviors of the metallic oxides.
The topic is important but not new. The possibility of performing in-situ characterization is interesting, but the authors could better explain the novelty of the results. For example, why is important to study the long term adsorption for this type of fast sensors? What improvements does this study make compared to the state of the art?
The manuscript presents several problems. Some methods and instruments are not described in sufficient detail to allow other researchers to reproduce the results. The measurement protocols are often unclear, mainly because the article frequently refers to previous works, making the reading neither smooth nor coherent. For example, the method used to measure long-term adsorption is unclear (is it estimated by the composition of the NO2 gas at the outlet or through the photoluminescence of the metal oxide?), the description of its measure protocol is poor of details as well as that of the results. Also, the main text of the paper is sometimes ambiguous or not very accurate, the terms in some equation are lost or not described and the same happens for some acronyms or symbol used; in many figures the axis header lacks details on the measured quantity or its measurement unit is lost. Furthermore, many figures are merged into a single figure making them difficult to see and their captions are poor in description. All these factors make the paper not fluid and difficult to the understanding for the reader. Finally large part of the references is not recent. Consequently, a thorough revision of the manuscript would be necessary, providing more explanations and details to make the text more fluid and clear, and highlighting the novelties while emphasizing the comparison with existing results in the literature, updating the literature itself.
For all the mentioned reasons the paper should be rejected.
Comments on the Quality of English LanguageThe manuscript is not fluid and more comments and information should be added.
Round 2
Reviewer 2 Report
Comments and Suggestions for Authors
The author has answered all the questions, but there are still some images with non-standard formats. Figures 2, 4, 6, 9, 10, 11, 13, and 15, please use all bounding boxes instead of half block diagrams.
Author Response
Figures 2, 4, 6, 9, 10, 11, 13, and 15 have been corrected in accordance with the reviewer's comment.